# The Interaction of the Endocannabinoid Anandamide and Paracannabinoid Lysophosphatidylinositol during Cell Death Induction in Human Breast Cancer Cells

**DOI:** 10.3390/ijms25042271

**Published:** 2024-02-14

**Authors:** Mikhail G. Akimov, Natalia M. Gretskaya, Evgenia I. Gorbacheva, Nisreen Khadour, Valeria S. Chernavskaya, Galina D. Sherstyanykh, Tatiana F. Kovaleko, Elena V. Fomina-Ageeva, Vladimir V. Bezuglov

**Affiliations:** Shemyakin-Ovchinnikov Institute of Bioorganic Chemistry, Russian Academy of Sciences, Miklukho-Maklaya 16/10, 117997 Moscow, Russia; natalia.gretskaya@gmail.com (N.M.G.); gorevig2000@gmail.com (E.I.G.); nisreenkhadour75@gmail.com (N.K.); lerachertlt2002@gmail.com (V.S.C.); galya24may@gmail.com (G.D.S.); tatianasolaris1@gmail.com (T.F.K.); evfa57@gmail.com (E.V.F.-A.); vvbez@ibch.ru (V.V.B.)

**Keywords:** CB1, CB2, GPR18, TRPV1, LPI, anandamide, LPI–AEA interaction, breast cancer

## Abstract

Endocannabinoid anandamide (AEA) and paracannabinoid lysophosphatidylinositol (LPI) play a significant role in cancer cell proliferation regulation. While anandamide inhibits the proliferation of cancer cells, LPI is known as a cancer stimulant. Despite the known endocannabinoid receptor crosstalk and simultaneous presence in the cancer microenvironment of both molecules, their combined activity has never been studied. We evaluated the effect of LPI on the AEA activity in six human breast cancer cell lines of different carcinogenicity (MCF-10A, MCF-7, BT-474, BT-20, SK-BR-3, MDA-MB-231) using resazurin and LDH tests after a 72 h incubation. AEA exerted both anti-proliferative and cytotoxic activity with EC_50_ in the range from 31 to 80 µM. LPI did not significantly affect the cell viability. Depending on the cell line, the response to the LPI–AEA combination varied from a decrease in AEA cytotoxicity to an increase in it. Based on the inhibitor analysis of the endocannabinoid receptor panel, we showed that for the former effect, an active GPR18 receptor was required and for the latter, an active CB2 receptor. The data obtained for the first time are important for the understanding the manner by which endocannabinoid receptor ligands acting simultaneously can modulate cancer growth at different stages.

## 1. Introduction

Initially, the endocannabinoid system included the ligands of cannabinoid receptors CB1 and CB2, anandamide (AEA) and 2-arachidonoylglycerol, found in the human body, and the enzymes of their biosynthesis and hydrolysis [1]. As the understanding of active molecules that can produce similar effects has expanded, other ligands have been identified, such as N-acyldopamines or N-acylglycines, as well as other receptors that mediate their effects, such as GPR55 and GPR18 [2]. It turned out that for some receptors of such an extended endocannabinoid system, the body contains ligands related to biologically active lipids that do not interact with CB1 and CB2 receptors. The name paracannabinoids has been proposed for these substances [3]. Lysophosphatidylinositol (LPI) is undoubtedly such a paracannabinoid, which is considered as an endogenous ligand of the non-classical cannabinoid receptor GPR55 but does not activate classical cannabinoid receptors [2].

At present, the so-called endocannabinoid system comprises classical (CB1, CB2, and TRPV1) and non-classical receptors (GPR55, GPR119, GPR92, and GPR18), metabolic enzymes (NAPE-PLD, FAAH, NAAA and MAGL), and several other ligands (2-arachidonoyl glycerol, N-arachidonoyl dopamine, and others which interact with the non-classical receptor subset) [4]. The classical cannabinoid receptors CB1 and CB2 are activated mainly by AEA and 2-arachidonoylglycerol, accordingly [4]. The non-classical cannabinoid receptor GRP55 is activated by two agonists, lysophosphatidylinositol (LPI) [5] and anandamide [6]. The ligand for GPR18 is N-arachidonoyl glycine, which arises metabolically from AEA [7,8,9]. The ligands of the endocannabinoid system are lipophilic molecules, which are usually synthesized and secreted on demand and act in an autocrine or paracrine fashion. Overall, the system modulates many processes in the range from memory and pain control to cancer cell death and proliferation.

Endocannabinoid concentrations and expression levels of cannabinoid receptors and enzymes responsible for endocannabinoid metabolism are usually associated with cancer aggressiveness. CB1 is found in 28% of breast carcinomas, with predominance in HER2^+^ tumors (14%), whereas CB2 is found in 72% of breast tumors, where it is again expressed predominantly in the HER2^+^ subtype (91%) [10]. Overexpression of CB2 is associated with a poor prognosis. Thus, a correlation was established between CB2 expression and tumor aggressiveness, since CB2 mRNA levels were higher in ER^-^/PR^-^ tumors than in ER^+^/PR^+^ tumors, as well as in HER2^+^ tumors than in HER2^-^ tumors, and in tumors with high histological tumors of low malignancy than in histological tumors of low malignancy. On the other hand, CB2 expression in ER^+^ and ER^-^ tumors is associated with a better prognosis. Gene expression data in various cancers show that for most of the cannabinoid receptors, there is a substantial correlation with the patient survival in the first ten years of the disease [11].

In the case of ligands and enzymes, in breast cancer, there is an increase in the level of N-acylphosphatidylethanolamine, a precursor of AEA, and MAGL (an enzyme that degrades 2-arachidonoyl glycerol) levels, mainly in ductal carcinomas [12]. In addition, LPI–GRP55 signal transmission is important for the modulation of migration and orientation of MDA-MB231 and MCF-7 cells [13]. Also, in basal-like and TNBC breast cancer cells, an increase in GRP55 expression correlates with more intense metastasis and a poor prognosis [14].

Endocannabinoids were shown to inhibit the growth of human breast and prostate cancer cells. It has been suggested that estrogen receptor-negative (ER^-^) cells are more sensitive to cannabinoids than estrogen receptor-positive (ER^+^) cells [12]. AEA inhibits the proliferation of human breast cancer cells by blocking the G0/G1–S phase transition of the cell cycle by interfering with signal transmission events associated with the CB1 receptor [15]. In MCF-7adrr cells, AEA disrupts the course of the cell cycle and induces apoptotic cell death [12]. On the other hand, LPI and its receptor GPR55 play a role in the modulation of migration, orientation, and polarization of breast cancer cells in response to the tumor microenvironment [13].

Cannabinoid receptors not only function individually, but they also form heterodimers with altered signaling properties; the physiological role of such complexes, however, is not always clear. As such, Balenga et al. demonstrated that GPR55 and CB2R interact with each other’s signaling pathways at the level of small GTPases, such as Rac2 and Cdc42. As a result, this leads to cellular polarization and effective migration, as well as to the abolition of degranulation and the formation of ROS in neutrophils [16]. Selective CB2 receptor agonists HU-308, HU-433, and HU-910 do not contribute to GPR55-mediated phosphorylation of ERK1/2 up to a concentration of 3 microns. However, the phosphorylation of ERK1/2 induced by LPI is inhibited by the (-)-enantiomer HU-308, whereas HU-308 does not affect the activity of LPI. The carboxyl analog of HU-910 strongly inhibits LPI-induced phosphorylation of ERK1/2 [17].

The GPR55 and CB1 receptors also can form heterodimers, and the co-expression of CB1 receptors in cells that stably express GPR55 specifically inhibits GPR55-mediated activation of transcription factors such as NFAT and SRE, as well as activation of kinase regulated by extracellular signal (ERK1/2). In addition, the presence of GPR55 enhances CB1R-mediated activation of ERK1/2 and NFAT [18].

Therefore, when the signals of AEA and LPI interact, the net outcome could be not a simple additivity of the effects but both an increase and a decrease in the individual activity. As such, for the combination of the endocannabinoid N-docosahexaenoyl dopamine (predominantly cytotoxic activity) with LPI (pro-proliferative or pro-migratory activity), we unexpectedly observed an increase in cytotoxicity [19]. A detailed understanding of the properties and mechanism of interactions of this kind is missing and could be important both for the development of the full picture of the endocannabinoid system’s participation in the cancer process and for cancer treatment.

In this paper, we, for the first time, evaluated the effect of LPI on AEA activity in six human breast cancer cell lines of different tumor degrees. Depending on the cell line, the response was in the range from a decrease in AEA cytotoxicity to an increase in it. For the former effect, an active GPR18 receptor was required, and for the latter, an active CB2 receptor.

## 2. Results

### 2.1. Cell Lines

The topic of this research was the direction and mechanism of the combined activity of the bioactive lipids AEA and LPI. It was logical to assume that at least the direction and magnitude of this interaction could depend on the grade of a tumor, as the cells progressively accumulate mutations and their response to various stimuli changes. To address this possibility, a panel of human breast cancer cell lines with different properties were used (Table 1). These lines were chosen to represent all of the major breast cancer subtypes [20,21]. 

To increase the selectivity of AEA toward the CB1 receptor, its chlorinated analog N-(2-chloroethyl)-5Z,8Z,11Z,14Z-eicosatetraenamide (ACEA)—a synthetic agonist of CB1 [22]—was used (Figure 1). This compound is widely used as an AEA replacement in various experiments [23,24,25].

**Table 1 ijms-25-02271-t001:** Subtypes and major receptor expression patterns in the cell lines used in this research [20,21].

Cell Line						
	MCF-10A	MCF-7	BT-474	SK-BR-3	BT-20	MDA-MB-231
Grade	Non-transformed	Luminal A	Luminal B	HER2	Basal	Triple-negative
Metastasis	−	Non-invasive [26]	Low [27]	Very low [27]	Low	High
ER	−	+	+	−	−	−
PR	−	+	+	−	−	−
HER2	−	−	+	+	−	−

### 2.2. Individual Effects of AEA and LPI on the Cell Lines

The first stage of our research was to determine the individual activity of the bioactive lipids AEA, ACEA, and LPI for the chosen cell lines. As far as AEA is known to exert anti-proliferative effects [28], and because LPI can act as a proliferation stimulator, the incubation time was chosen to be 72 h. Two cell viability detection methods were used: a resazurin test for viable cells and a lactate dehydrogenase test for dead cells. The concentration range for the activity testing was chosen based on data from the literature. For AEA, cytotoxicity effects could be observed up to 100 µM [29,30], after which problems with specificity and solubility might occur. For LPI, the standard concentration range is up to 10 µM [31,32].

For all cell lines tested, AEA exerted both anti-proliferative (Figure 2) and cytotoxic (Figure 3) activity with EC_50_ in the range from 31 to 80 µM (Table 2 and Table 3). LPI did not exert any significant effect on any of the cell lines. For SK-BR-3, the EC_50_ value in the LDH test was higher than in the resazurin test, indicating a consecutive anti-proliferative and cytotoxic action. For BT-20 and MDA-MB-231, an opposite situation was observed.

As an anti-proliferative agent, ACEA was more active than AEA only for the BT-474 cell line, while it showed diminished activity on SK-BR-3 cell lines (Table 2). For the last cell line, ACEA demonstrated increased cytotoxic activity (Table 3). But in general, the observed activity of the ACEA compound was close to or slightly lower than the one of AEA, so we chose AEA for further combinations study.

### 2.3. The Effect of LPI–AEA Combinations on the Viability of the Cell Lines

We next tested the ability of LPI to change the action of AEA when both substances were added together. We used several AEA and LPI concentrations to account for the possible activity changes in different concentration ranges. In this experiment series, we also used a 72 h incubation with the substances to detect possible proliferation changes and applied both resazurin and LDH tests to differentiate between the anti-proliferative and anti-cytotoxic effects.

By the effect of LPI addition, the cell lines could be split into three groups (Figure 4 and Figure 5):No effect of LPI on AEA activity (BT-20);LPI decreases AEA’s anti-proliferative effect (BT-474);LPI increases AEA’s anti-proliferative and cytotoxic effects (MCF-10A, MCF-7, SK-BR-3, MDA-MB-231).

In most cases, the response in the LDH test reflected the one in the resazurin test, indicating a simultaneous change in the cytotoxic and anti-proliferative effects. However, in the MDA-MB-231 and SK-BR-3 cell lines, only the anti-proliferative component was reduced. A similar effect was observed for the BT-474 cell line, in which LPI addition decreased the effect of AEA.

### 2.4. Receptor Participation in the Individual and Combined Substance Effects

The next goal of our research was determining the mechanism of the LPI effect on AEA activity. We hypothesized that LPI–AEA interaction could proceed through one of the known cannabinoid receptors, as some of them (CB1, CB2, TRPV1, GPR55) are targets for AEA [33], others are targets for LPI (GPR55) [34], and at least for some of them, a formation of activity-changing heterodimers was described (GPR18) [35,36,37]. 

We first evaluated the expression of the core cannabinoid receptors (CB1, CB2, and GPR55) in the model cell lines using qPCR and Western blotting (Figure 6 and Figure 7, Appendix A). CB1 *(CNR1)* mRNA levels were about 200 times lower in BT-474 and BT20 cell lines, compared to others. The expression of CB2 *(CNR2)* was similar in all the cell lines, except for BT20, where it was 4 times higher. In MCF-10A and SK-BR-3, *GPR55* gene expression was negligible, while in the other 4 cell lines, it was approximately similar. However, on the Western blots, the GPR55 band was visible for the MCF-10A cell line. This discrepancy could be explained by a slow receptor turnover in these cells, which could pose much lesser requirement for the mRNA levels. *GPR18* receptor gene expression was present in all cell lines, varying 4-fold among them. The *TRPV1* receptor mRNA was present in all cell lines on a similar level. Hierarchical clustering of the receptor expression data demonstrated that MCF7 and MDA-MB-231 are the most similar ones, and then the similarity decreased in the following order: (MDA-MB-231~MCF7) > SK-BR-3 > MCF-10A > BT20 > BT-474.

Next, we added a selective blocker for each of those receptors, both to AEA alone and to the AEA–LPI combination to check for the importance of the appropriate receptor, and evaluated the proliferation change in the resazurin test after 72 h of incubation with the cells. However, in some cases, neither receptor blocker removed the effect of LPI, and we added a blocker of GPR18, a second receptor for the LPI [34], to the panel.

The effect of receptor blockers differed substantially between the cell lines (Figure 8). In BT-20, the only active blocker was the one of CB1, and its effect was very close both in the presence and in the absence of LPI. A similar effect of this blocker was also observed in MCF-7, BT-474, and MDA-MB231. In BT-474, the GPR18 blocker removed or substantially decreased the effect of LPI addition. In MCF-7 and MDA-MB-231, the blocker of CB2 counteracted the LPI effect, and the GPR18 blocker did not change the substance activity. In MCF-10A and SK-BR-3, neither blocker was active.

## 3. Discussion

In this paper, we studied the interaction of AEA and LPI in a panel of human breast cancer cell lines of different tumor grades. The possibility of such interaction was demonstrated by us in a previous work, in which LPI increased the cytotoxicity of another endocannabinoid N-docosahexaenoyl dopamine on the MDA-MB-231 cell line [19]. This interaction could be quite important in the cancer setting, as the endocannabinoid system is a possible target for anti-cancer therapy [38], and LPI is usually considered a pro-proliferative molecule [14]. We found that depending on the receptor set of the cell line, AEA’s anti-proliferative effect could be enhanced, counteracted, or not affected by LPI’s presence.

In all cell lines tested, AEA had either a cytotoxic effect or a mixture of cytotoxic and anti-proliferative ones. This observation was in line with the already published data [39,40], and an anti-proliferative effect was also observed at low concentrations of the substance [41]. Our research, however, extends the known data to several cell lines, namely BT-20, MCF-10A, and SK-BR-3, and adds data on the activity of selective CB1 receptor agonist ACEA in these models. ACEA behaves in cell systems close to AEA and widely used in endocannabinoid research [23,24,25]. The observed difference between AEA and ACEA EC_50_ is of particular interest. The substitution of the hydroxyl group by chlorine does not prevent ACEA interaction with cannabinoid receptors [42] but shifts its selectivity towards CB1 [22]. However, it probably prevents its direct metabolism to N-arachidonoyl glycine [9] and consequent interaction with the GPR18 receptor, which recognizes only the latter molecule [43]; the observed difference of the AEA effect could be a result of simultaneous activation of both of these receptors. This hypothesis is in line with the importance of the GPR18 receptor for LPI–AEA interaction, also described in this paper. In our experiments, we did not observe relevant differences between AEA and ACEA in effects on the viability of cancer cell lines.

Contrary to our expectations, LPI had different effects on AEA activity depending on the cell line, ranging from pro-cytotoxicity to pro-proliferation. The pro-cytotoxic effect was previously described by us for the N-docosahexaenoyl dopamine–LPI pair [19]. Of particular interest was the fact that the mode of the LPI action mostly depended on the receptor set of the cell line and not on the tumor grade. This could have important implications for the potential therapeutic use of endocannabinoid system ligands and inhibitors, as it shows the possibility of how the same treatment could potentially lead to opposite effects.

The experiments of the cannabinoid receptor blockers’ influence on LPI–AEA interaction demonstrated the importance of two of them: CB2 in MCF-7 and MDA-MB-231 and GPR18 in MCF-10, BT-474, and SK-BR-3. The effect of AEA was substantially suppressed by the CB1 receptor blocker in the BT-474, BT-20, MDA-MB-231, and, to some extent, MCF-7 cell lines, while in MCF-10 and SK-BR-3, the effect of the receptor blockers was negligible.

LPI does not significantly recruit β-arrestin at GPR18, and LPI GPR18 activity through other signaling pathways has not yet been reported; it also does not interact with the CB2 receptor [44]. On the other hand, LPI directly activates RhoA [45], and CB2 receptor activation also elicits RhoA this effect [46]. Thus, the cases of the CB2 requirement for LPI’s pro-cytotoxic action could be attributed to the co-activation of RhoA by both pathways. 

The requirement of GPR18 for LPI to counteract AEA could be due to a form of indirect interaction. Both GPR55 (a target of LPI) [35] and GPR18 [36] form cross-inhibitory heterodimers with CB2; in addition, GPR18 activation is pro-proliferative [47]. Thus, when GPR18 is active, its activation by AEA (directly or after a metabolic activation [7,8]) could shift the overall response to be less cytotoxic, and an additional inhibition of CB2 via the cross-talk with the LPI-activated GPR55 could enhance the effect.

The observed effects, especially on MCF-10A and SK-BR-3 cell lines, could also be realized through the participation of the additional receptor targets of LPI. Based on data from the literature, beyond GPR55, this substance can activate TRPV2 [48], RhoA [45], and GPR119 [49].

The detected ability of LPI to both increase and decrease AEA activity could be important for the understanding of the endocannabinoid modulation of cancer growth at different stages. In the context of cancer treatment with the endocannabinoid system as a target, it could be important to consider the state of the LPI–AEA interaction mode at the given disease stage, as it could direct the usability of an application of LPI synthesis inhibitors or signaling blockers. The exact molecular mechanism of this interaction requires further research. Of particular interest could be the cases of MCF-10A and SK-BR-3 cell lines, in which none of the CB1, CB2, GPR55, or GPR18 receptors were involved in AEA activity and AEA–LPI interaction.

## 4. Materials and Methods

### 4.1. Reagents and Cell Lines

DMEM/F12, DMEM, IMDM, L-glutamine, HCl, Earle’s salts solution, Hank’s salts solution, Versene’s solution, antibiotic/antimycotic mixture (penicillin, streptomycin, amphotericin B), DMEM, trypsin, and fetal bovine serum were from Servicebio, Hubei, China.

Cell lines MDA-MB-231 (HTB-26), MCF-10A (CRL-10317), MCF-7 (HTB-22), BT-474 (HTB-20), BT-20 (HTB-19), HL-60 (CCL-240), LN-229 (CRL-2611), and SK-BR-3 (HTB-30) were purchased from ATCC, Manassas, VA, USA.

Antibodies anti-b-actin, anti-CB1, anti-CB2, and anti-GPR55 were from Abcam, Cambridge, UK. Anti-mouse IgG antibody was from Jackson ImmunoResearch, Cambridge, UK.

SR 144028, PSB CB5, ML-184, ML-193, and LPI were from Tocris Bioscience, Bristol, UK. DMSO, resazurin, NAD^+^, DL-lactate, diaphorase, iodonitrotetrazolium chloride (INT), D-glucose, glycylglycine, acetic acid, MgSO_4,_ EGTA, dithiothreitol, DMSO, Triton X-100, acrylamide, bis-acrylamide, Triton X-100, SDS, nitro blue tetrazolium, Tris, EDTA, agarose, bicinchoninic acid, D-glucose, bovine serum albumin, anti-rabbit IgG antibody, and 5-Bromo-4-chloro-3-indolyl phosphate-toluidine were from Sigma-Aldrich, St. Louis, MO, USA. The purity of all used reagents was 95% or more.

rhEGF was from SciStoreLab, Skolkovo, Russia.

### 4.2. Chemical Synthesis

Anandamide and ACEA were synthesized as described previously [50]. In short, anandamide (AEA) and chloro-anandamide (ACEA, (5Z,8Z,11Z,14Z)-N-(2-chloroethyl)icosa-5,8,11,14-tetraenamide) were synthesized from arachidonic acid and ethanolamine or 2-chloroethylamine, respectively, after activation of the carboxyl group of arachidonic acid with isobutyl chloroformate in the presence of triethylamine. AEA’s physico-chemical constants were in accordance with published data. For ACEA: ^1^H-NMR (δ, CDCl_3_: 0.902 [3H, H20], 1.313 [8H, H19,18,17,16], 1.745 [3H, H3], 2.080 [4H, H4], 2.232 [2H, H2], 2.825 [6H, H7,10,13,], 2.633 [4H, H 2’, 1’], 5.387 [8H, -CH=CH-], 5.852[1H, NH]. ESI MS m/z (C_22_H_36_ClNO, exact mass: 365.2485): 366.2335 [M+H]^+^, 388.2335 [M+Na]^+^, 404.2061 [M+K]^+^

### 4.3. Cell Culture

Cells were cultured in DMEM (cell lines MDA-MB-231, BT-474, MCF-7, and BT-20, LN-229), IMDM (HL-60), or McCoy 5A (cell line SK-BR-3) supplemented with 10% FBS, 4 mM L-glutamine, 100 U/ml penicillin, 100 μg/ml streptomycin, and 2.5 μg/ml amphotericin B. The medium for MCF-10A consisted of DMEM/F12 supplemented with 10% FBS, 4 mM L-glutamine, 100 U/ml penicillin, 100 μg/ml streptomycin, 2.5 μg/ml amphotericin B, 20 ng/ml EGF, 0.5 µg/ml hydrocortisone, 10 μg/ml insulin, and 100 nM (−)-isoproterenol [51]. All the cells were maintained at 5% CO_2_ and 100% humidity at 37°C. Cells were subcultured by successive treatment with Versene’s solution and trypsin solution. Mycoplasma contamination was controlled using the Jena Biosciences (Jena Biosciences, Jena, Germany) kit according to the manufacturer’s instructions.

### 4.4. Cytotoxicity and Proliferation Evaluation

Cells were seeded in 96-well plates (2000 per well) in a volume of 100 µl of medium and cultured for a day. After that, the substance was added at the required concentration in the range from 0.01 to 150 μM in the form of a DMSO solution in a fresh 100 μl of the medium, in which the serum was replaced with a delipidated analog; the final DMSO concentration was 0.5% or less. If receptor blockers were used, they were added to 50 μl of the medium one hour before the addition of the substance and then the second portion (to ensure the constancy of concentration) with the substance; the volume of the medium in which the substance was added was 50 µl in this case. For the proliferation induction studies, the medium with the delipidated FBS was used to eliminate the influence of the natural LPI. The cells were incubated with the substance for 72 h, after which the viability was determined using the resazurin test and the cell death using the LDH test. Each experiment was repeated at least five times.

### 4.5. Western Blotting

To evaluate the expression of particular proteins in the cells, the cells were seeded at a density of 200,000 per well in a 24-well plate the day before the experiment. After that, the cells were washed once with PBS, lysed using the lysis solution (150 mM NaCl, 1% Triton X-100, 0.1% SDS, 50 mM Tris-HCl pH 8.0, 1% protease inhibitor cocktail) for 30 min at +4 °C, and centrifuged for 5 min at 10,000× *g*. The total protein concentration in the supernatants was determined using the BCA assay. Proteins were separated using denaturing SDS-PAGE in 10% gel with the PageRuler protein ladder (Thermo Fisher Scientific, Waltham, MA USA), transferred to a nitrocellulose membrane using the Invitrogen Power Blotter with the Invitrogen Power Blotter 1-step transfer buffer (Thermo Fisher Scientific, Waltham, MA USA) and Invitrogen precut membranes and filters (Thermo Fisher Scientific, Waltham, MA USA), and stained with antibodies using the Invitrogen iBind system (Thermo Fisher Scientific, Waltham, MA USA) according to the manufacturer’s protocol. The following antibodies were used: rabbit anti-GPR55 (Abcam ab203663), rabbit anti-CB2 (Abcam ab45942), rabbit anti-CB1 (Abcam ab23703), and mouse anti-beta-actin (Abcam ab8226); and secondary antibodies (coupled to alkaline phosphatase) anti-rabbit IgG (Sigma-Aldrich A9919) and anti-mouse IgG (Jackson ImmunoResearch 111-055-003). After the staining, the membrane was washed in H_2_O for 10 min and incubated with the staining solution (20 µl BCIP solution + 30 µl NBT solution per 10 ml of substrate buffer) for 1 h at room temperature. Substrate buffer for alkaline phosphatase: 100 mM Tris-HCl, pH 9.5, 100 mM NaCl, and 5 mM MgCl_2_. BCIP solution: 20 mg/ml 5-Bromo-4-chloro-3-indolyl phosphate-toluidine (BCIP) in 100% dimethyl formamide. NBT staining solution: 50 mg/ml nitro blue tetrazolium (NBT) in 70% dimethyl formamide.

### 4.6. RNA Isolation and cDNA Synthesis

RNA was isolated from in vitro cultivated cells using the QIAzol Lysis Reagent (Qiagen, Hilden, Germany) as described in manufacturer’s protocol. RNA samples were treated with DNase I (Thermo Fisher Scientific, Waltham, MA USA) according to the manufacturer’s instruction. The RNA concentration was measured using a Nanodrop OneC Spectrophotometer (Thermo Fisher Scientific, Waltham, MA USA). cDNA was synthesized with the MMLV RT kit (Evrogen, Moscow, Russia) according the manufacturer’s recommendations.

### 4.7. qPCR

qPCR was performed on a LightCycler 96 (Roche, Basel, Switzerland) with qPCRmix-HS SYBR reagent (Evrogen, Moscow, Russia). Cycling conditions were 95 °C for 150 s and then 45 cycles of 95 °C for 20 s, 57 °C for 20 s, and 72 °C for 20 s. Primer specificity was confirmed by visualizing DNA on an agarose gel following PCR. The relative level of expression was determined by the 2-ΔΔCt method. *beta-2 microglobulin and POL2R* were used as an internal controls. Each analysis was performed in triplicate. Primer sequences for PCR and qPCR were as follows: *beta-2 microglobulin* forward 5’-CAGCAAGGACTGGTCTTTCTAT-3’, reverse 5’-ACATGTCTCGATCCCACTTAAC-3’, *CB1 (CNR1)* forward 5’-CAAGCCTCTCTGGCACTTT-3’, reverse 5’-CTGGTGGTTGGGCCTATTT-3’, *CB2 (CNR2)* forward 5’-CTACACCTATGGGCATGTTCTC-3’, reverse 5’-CCTCACATCCAGCCTCATTC, *GPR55* forward 5’-ACTGATGTGCTTCCCTTTGAT-3’, reverse 5’-CCTGAACACTGGGTGGTATAAG-3’, *GPR18* forward 5’-GTGTGTGGGAGTCTGGATAATG-3’, reverse 5’-GTCAGAAATCTTGAGGCAGGT-3’, *TRPV1* forward 5’-CTGGACCAACATGCTCTACTAC-3’, reverse 5’-AGGTCTCTCAGGATCATCTTCT-3’, *POL2R* forward 5’-CCCAGCTCCGTTGTACATAAA-3’, reverse 5’-TCTAACAGCACAAGTGGAGAAC-3’.

### 4.8. BCA Protein Assay

Protein concentration was determined using the BCA assay [52]. The following base reagents were used: Reagent A (bicinchoninic acid 1%, Na_2_CO_3_*H_2_O 2%, sodium tartrate 0.16%, NaOH 0.4%, NaHCO_3_ 0.95%, pH 11.25), Reagent B (4% CuSO_4_*5H_2_O), and S-WR (50 volumes of Reagent A + 1 volume of Reagent B). An amount of 5 µl of cell lysate was mixed with 40 µl of S-WR and incubated for 15 min at 60 °C, after which the optical density was measured at λ = 562 nm using the Hidex Sense Beta Plus microplate reader (Hidex, Turku, Finland). Each sample was assayed in triplicate. Cell lysis buffer was used as a background control. Bovine serum albumin solution in the cell lysis buffer was used as a positive control and to build a calibration curve.

### 4.9. Resazurin Test

To evaluate cell viability, the culture medium in the wells was replaced with a 0.2 mM resazurin solution in Earle’s solution with the addition of 1 g/l D-glucose and incubated for 1.5 h at 37 °C under cell culture conditions [53]. After that, the fluorescence of the solution was determined at the excitation wavelength of 550 nm and the emission wavelength of 590 nm using the Hidex Sense Beta Plus microplate reader (Hidex, Turku, Finland). The positive control was the cell culture treated with the solvent alone, and the negative control was treated with 0.9% Triton X-100.

### 4.10. LDH Test

To evaluate cell death, an assay for the activity of the intracellular enzyme lactate dehydrogenase released into the medium from the dead cells was used [54]. To this end, a 75 µl aliquot of the culture medium from each well was transferred to a fresh 96-well plate. After that, 10 µl of the following reagents were added to each well: 36 mg/ml of lactate in phosphate-buffered saline, pH 7.2; 2 mg/ml of INT in the diaphorase buffer (see below); 3 mg/ml of NAD^+^ mixed with 6 U/ml diaphorase in 0.03% bovine serum albumin and 1.2% sucrose in phosphate-buffered saline, pH 7.2. The reaction mixture was incubated for 20 min at room temperature, and the optical density of the solutions was determined at the wavelength of 490 nm using the Hidex Sense Beta Plus microplate reader (Hidex, Turku, Finland). The positive control was the cell culture treated with the solvent alone, and the negative control was treated with 0.9% Triton X-100. 

### 4.11. Statistical Analysis

Statistical evaluation was performed using GraphPad Prism 9.3 software (https://www.graphpad.com/, accessed on 10 December 2023) and using the R Statistical Language (https://www.r-project.org/, accessed on 10 December 2023). ANOVA with the Holm–Sidak or Dunnett post-test was used to compare the obtained values; *p* ≤ 0.05 was considered significant.

## 5. Conclusions

Based on the obtained data, LPI can substantially and bidirectionally change the activity of AEA depending on the cell signaling context. A typical response is an increase in AEA cytotoxicity with the participation of the CB2 receptor. However, in some cases, LPI’s presence enhances GPR18 signaling, leading to a decrease in AEA-induced cell death. This response type switch could be important for the understanding of the endocannabinoid modulation of cancer growth at different stages. In the context of cancer treatment with the endocannabinoid system as a target, it could be important to take into account the state of the LPI–AEA interaction mode at the given disease stage, as it could direct the usability of an application of LPI synthesis inhibitors or signaling blockers.

## Figures and Tables

**Figure 1 ijms-25-02271-f001:**
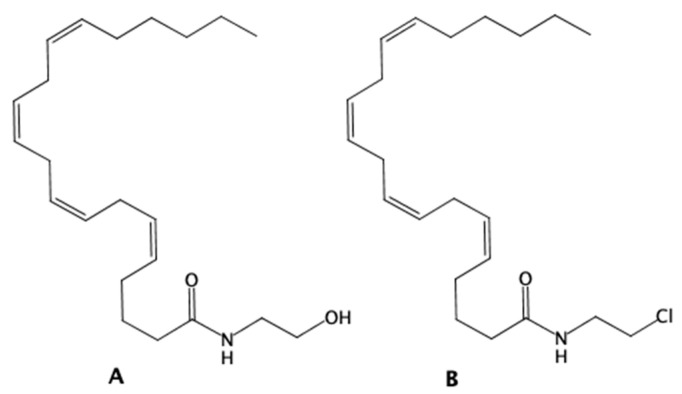
The structure of anadamide (AEA) (**A**) and N-(2-chloroethyl)-5Z,8Z,11Z,14Z-eicosatetraenamide (ACEA) (**B**).

**Figure 2 ijms-25-02271-f002:**
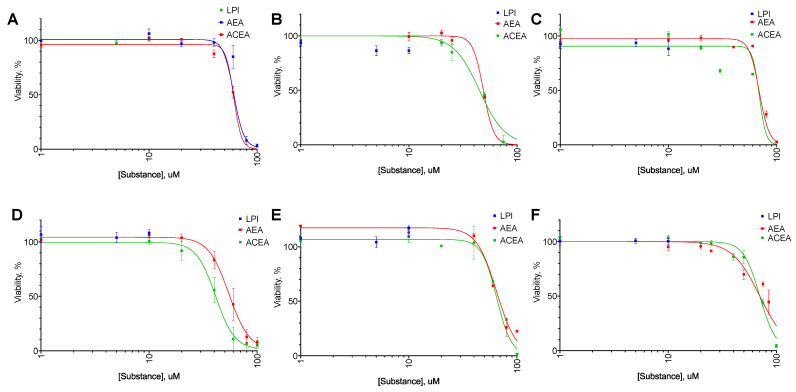
The anti-proliferative effect of AEA, ACEA, and lysophosphatidylinositol (LPI) on breast cancer cell lines. Incubation time: 72 h; resazurin test, mean ± standard error (*n* = 4 experiments). (**A**) MCF-10A, (**B**) MCF-7, (**C**) BT-474, (**D**) SK-BR-3, (**E**) BT-20, (**F**) MDA-MB-231.

**Figure 3 ijms-25-02271-f003:**
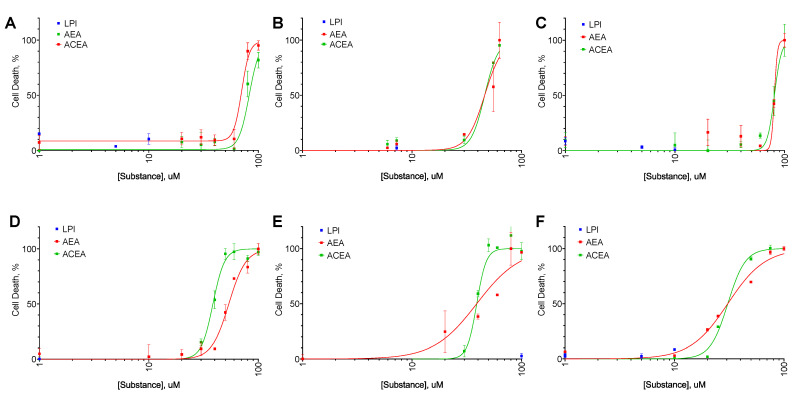
The effect of AEA, ACEA, and LPI on breast cancer cell lines cell death. Incubation time: 72 h; LDH test, mean ± standard error (*n* = 4 experiments). (**A**) MCF-10A, (**B**) MCF-7, (**C**) BT-474, (**D**) SK-BR-3, (**E**) BT-20, (**F**) MDA-MB-231.

**Figure 4 ijms-25-02271-f004:**
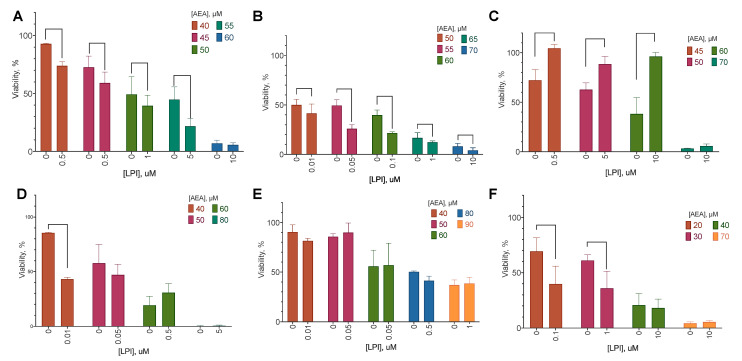
The effect of AEA combination with LPI on breast cancer cell lines’ viability. Incubation time: 72 h; resazurin test, mean ± standard error (*n* = 4 experiments). (**A**) MCF-10A, (**B**) MCF-7, (**C**) BT-474, (**D**) SK-BR-3, (**E**) BT-20, (**F**) MDA-MB-231.

**Figure 5 ijms-25-02271-f005:**
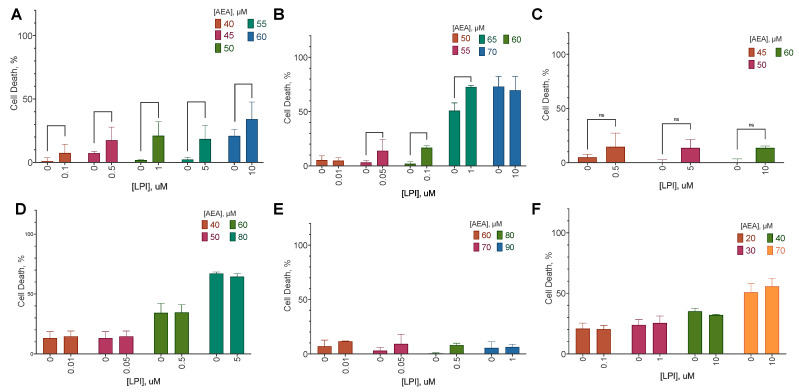
The effect of AEA combination with LPI on breast cancer cell lines’ cell death. Incubation time: 72 h; LDH test, mean ± standard error (*n* = 4 experiments). (**A**) MCF-10A, (**B**) MCF-7, (**C**) BT-474, (**D**) SK-BR-3, (**E**) BT-20, (**F**) MDA-MB-231.

**Figure 6 ijms-25-02271-f006:**
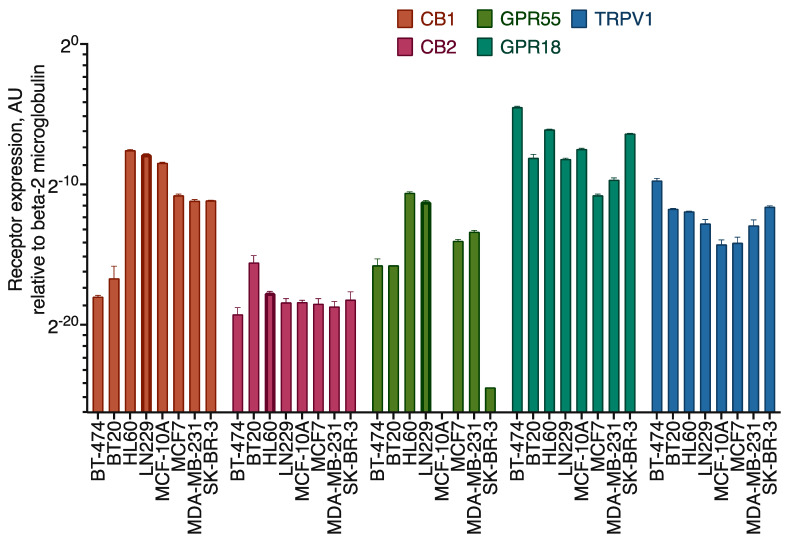
qPCR evaluation of the cannabinoid receptor expression in the model cell lines. HL60 and LN229 cell lines were used as a reference with a high expression of *CB2* and *CB1* and *GPR55* receptor pair, accordingly.

**Figure 7 ijms-25-02271-f007:**
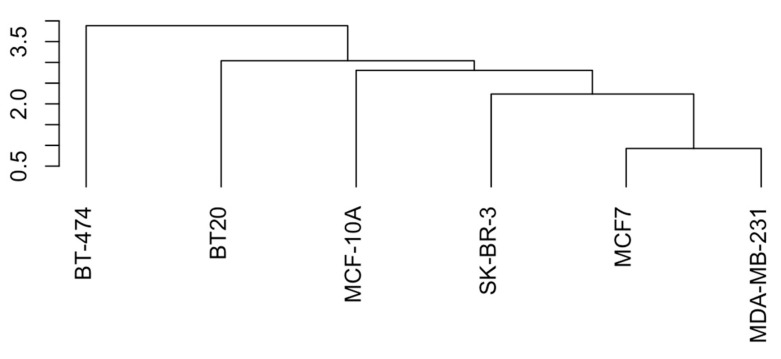
Dendrogram of the cannabinoid receptor mRNA expression similarity in the model cell lines.

**Figure 8 ijms-25-02271-f008:**
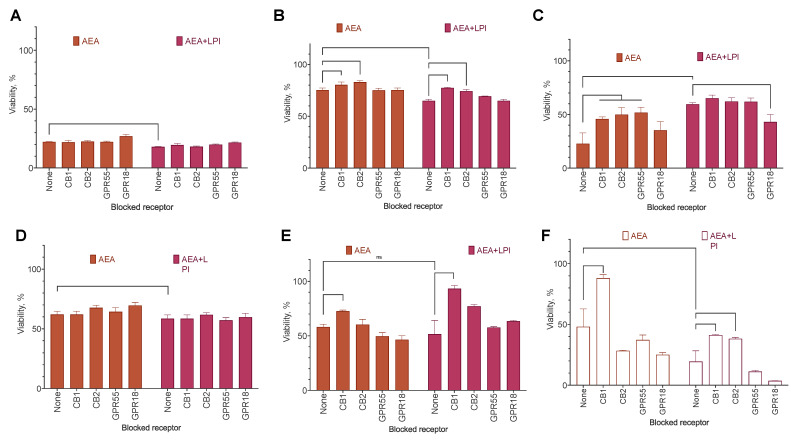
The effect of receptor blockers on the effect of AEA’s combination with LPI on breast cancer cell lines’ viability. The following substances and concentrations were used: CB1, SR 141716A (100 nM); CB2, SR 144528 (100 nM); GPR55, ML-193 (2 µM); GPR18, PSB CB5 (3 µM). Incubation time: 72 h; resazurin test, mean ± standard error (*n* = 4 experiments). (**A**) MCF-10A, (**B**) MCF-7, (**C**) BT-474, (**D**) SK-BR-3, (**E**) BT-20, (**F**) MDA-MB-231.

**Table 2 ijms-25-02271-t002:** The inhibitory effect of AEA, ACEA, and LPI on breast cancer cell lines’ proliferation. Incubation time: 72 h; resazurin test, mean ± standard error (*n* = 4 experiments).

	Cell Line					
	MCF-10A	MCF-7	BT-474	SK-BR-3	BT-20	MDA-MB-231
	EC_50_, µM, mean (95% CI)					
AEA	68.78 (63.18 to 74.87)	49.65 (38.35 to 67.15)	74.87 (71.99 to 77.87)	42.99 (38.65 to 47.81)	66.53 (61.91 to 71.49)	71.91 (63.80 to 79.36)
ACEA	61.18 (58.76 to 63.69)	50.43 (42.45 to 68.10)	45.57 * (34.18 to 60.75)	45.31 * (34.15 to 47.70)	58.64 (55.31 to 60.96)	68.99 (65.56 to 72.54)
LPI	>10	>10	>10	>10	>10	>10

*, a statistically significant difference between the AEA and ACEA EC_50_, i.e., *p* ≤ 0.05 in the ANOVA with the Holm–Sidak post-test.

**Table 3 ijms-25-02271-t003:** The effect of AEA, ACEA, and LPI on breast cancer cell lines’ cell death. Incubation time: 72 h; LDH test, mean ± standard error (*n* = 4 experiments).

	Cell Line					
	MCF-10A	MCF-7	BT-474	SK-BR-3	BT-20	MDA-MB-231
	EC_50_, µM, mean (95% CI)					
AEA	82.45 ** (70.97 to 95.77)	55.19 (46.21 to 56.61)	80.75 (76.30 to 85.45)	53.81 ** (50.14 to 57.74)	45.75 ** (35.75 to 55.04)	37.38 ** (30.78 to 45.40)
ACEA	75.87 ** (70.99 to 82.97	43.81 (41.60 to 52.30)	81.03 ** (75.77 to 86.61)	38.54 * (36.80 to 40.36)	38.90 *,** (36.72 to 41.22)	35.01 ** (32.89 to 37.27)
LPI	>10	>10	>10	>10	>10	>10

*, a statistically significant difference between the AEA and ACEA EC_50_, i.e., *p* ≤ 0.05 in the ANOVA with the Holm–Sidak post-test; **, a statistically significant difference from the EC_50_ value in the resazurin test (see Table 2), i.e., *p* ≤ 0.05 in the ANOVA with the Holm–Sidak post-test.

## Data Availability

The data presented in this study are available on request from the corresponding author. The data are not publicly available due to legal issues.

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
