# Peer review of "The Interaction of the Endocannabinoid Anandamide and Paracannabinoid Lysophosphatidylinositol during Cell Death Induction in Human Breast Cancer Cells"

_ijms, 2024, doi:10.3390/ijms25042271_

Round 1

Reviewer 1 Report

Comments and Suggestions for Authors

My comments

Title: The Interaction of the Endocannabinoid Anandamide and GPR55 Agonist Lysophosphatidylinositol during Cell Death Induction in Human Breast Cancer Cells

In this manuscript, the authors studied the synergetic effects of the Endocannabinoid Anandamide and GPR55 Agonist Lysophosphatidylinositol on Human Breast Cancer Cells. The manuscript is meaningful, but does not address research questions, is relevant in the field and does not address a specific gap. However, I have some major comments for the author.

  1. The manuscript's experimental design and analysis are very preliminary.
  2. The author should do proper statistical analysis and graphical representation.
  3. The author should provide an animal tumour model experiment with at least one cell to confirm this activity.
  4. There are more confirmational studies required. 
Comments on the Quality of English Language

 Moderate editing of English language required

Author Response

Point 1. Moderate editing of English language required

Response 1. English spell and grammar check was performed
Point 2. Introduction must be improved.

Response 2. The introduction was extended to provide more information on the relevance of cannabinoid receptors, their ligands and signaling for the breast cancer pathological process.

Point 3. Not all cited references are relevant to the research.

Response 3. The references list was checked for inappropriate ones.

Point 4. The research design must be improved. The manuscript's experimental design and analysis are very preliminary.

Response 4. The research design was updated with qPCR profiling of cannabinoid receptor expression in the model cell lines

Point 5. Method description must be improved.

Response 5. Method description was extended.

Point 6. The presentation of the results must be improved.

Response 6. The presentation and description of the results was extended.

Point 7. The conclusions must be improved.

Response 7. The conclusions were updated and made more precise.

Point 8. The manuscript does not address research questions, is relevant in the field and does not address a specific gap.

Response 8. The introduction was extended to describe the relevance and the specific gap.

Point 9. The author should do proper statistical analysis and graphical representation.

Response 9. Statistical evaluation was added. The graphical representation was updated to be more appropriate.

Point 10. The author should provide an animal tumour model experiment with at least one cell to confirm this activity.

Response 10. The aim of the study was to determine the interaction of the substances on the molecular and cellular level. The animal studies are, therefore, out of the scope of the research.

Point 11. There are more confirmational studies required. 

Response 11. qPCR evaluation of the gene expression was added.

Reviewer 2 Report

Comments and Suggestions for Authors

Reviewer Comments

The manuscript demonstrates that following Interaction of the Endocannabinoid Anandamide and GPR55 Agonist Lysophosphatidylinositol in Human Breast Cancer Cells. Are these proteins considered to be natural or canonical targets lysophosphatidylinositol, or is this enhanced turnover a specific consequence of the receptors?

1.       The authors state that CB1 is associated with about 25% of breast carcinomas, with no reference given. Estimates vary, but most recent estimates put the percentage of breast cancers that are CB1 to be significantly higher than 25%

2.       The authors note the death domain carried by the target proteins. What other proteins carry this same domain, and is it anticipated that some or all of them might also be targets for LPI?

3.       There was no statistical difference between the normal to AEA and ACEA fig 2. Another way to demonstrate specificity would be the use of viability test which is not done.

4.       The comparison between AEA and ACEA cancers for LDH test is less compelling because we do not know if these two types of cancers even originate from the same epithelial cell type, meaning that other reasons, not contemplated by the authors.

5.       The difference in expression of cannabinoid receptors comparing CB1 versus CB2 in breast tumors is relatively small and could be explained by a number of other factors. These should be included in interpreting the results.

Author Response

Point 1. The manuscript demonstrates that following Interaction of the Endocannabinoid Anandamide and GPR55 Agonist Lysophosphatidylinositol in Human Breast Cancer Cells. Are these proteins considered to be natural or canonical targets lysophosphatidylinositol, or is this enhanced turnover a specific consequence of the receptors?

Response 1. The protein GPR55 is considered to be a canonical target for lysophosphatidylinositol. Other receptors studied in the word (CB1 and CB2) are canonical targets for anandamide.

Point 2.  The authors state that CB1 is associated with about 25% of breast carcinomas, with no reference given. Estimates vary, but most recent estimates put the percentage of breast cancers that are CB1 to be significantly higher than 25%

Response 2. The estimates were updated, and the appropriate references were added.

Point 3. The authors note the death domain carried by the target proteins. What other proteins carry this same domain, and is it anticipated that some or all of them might also be targets for LPI?

Response 3. LPI is a selective ligand for GPR55 and GPR18 receptors.

Point 4.  There was no statistical difference between the normal to AEA and ACEA fig 2. Another way to demonstrate specificity would be the use of viability test which is not done.

Response 4. Both viability (resazurin) and cytotoxicity (LDH) test were done for all of the cell lines treated with the bioactive lipid molecules AEA and ACEA.

Point 5. The comparison between AEA and ACEA cancers for LDH test is less compelling because we do not know if these two types of cancers even originate from the same epithelial cell type, meaning that other reasons, not contemplated by the authors.

Response 5. AEA and ACEA are not cancers. They are bioactive lipid molecules, which interact with the cannabinoid receptors. Therefore, their comparison is appropriate.

Point 6.  The difference in expression of cannabinoid receptors comparing CB1 versus CB2 in breast tumors is relatively small and could be explained by a number of other factors. These should be included in interpreting the results.

Response 6. The studied compounds are not expected to affect the expression of the CB1 and CB2 receptors. Rather, they activate these receptors. That is why we performed the expression profiling of the model cell lines to make sure that they do contain the appropriate proteins. The substance activity did not substantially correlate with the differences in the CB1 and CB2 expression between the cell lines used.